# Diverse Begomoviruses Evolutionarily Hijack Plant Terpenoid-Based Defense to Promote Whitefly Performance

**DOI:** 10.3390/cells12010149

**Published:** 2022-12-30

**Authors:** Ning Wang, Pingzhi Zhao, Duan Wang, Muhammad Mubin, Rongxiang Fang, Jian Ye

**Affiliations:** 1State Key Laboratory of Plant Genomics, Institute of Microbiology, Chinese Academy of Sciences, Beijing 100101, China; 2CAS Center for Excellence in Biotic Interactions, University of Chinese Academy of Sciences, Beijing 100049, China; 3Virology Lab, CABB University of Agriculture, Jail Road, Faisalabad 38000, Pakistan

**Keywords:** begomovirus-betasatellite complex, indirect mutualism, whitefly vector, terpenoid, βC1 protein

## Abstract

Arthropod-borne pathogens and parasites are major threats to human health and global agriculture. They may directly or indirectly manipulate behaviors of arthropod vector for rapid transmission between hosts. The largest genus of plant viruses, *Begomovirus*, is transmitted exclusively by whitefly (*Bemisia tabaci*), a complex of at least 34 morphologically indistinguishable species. We have previously shown that plants infected with the tomato yellowleaf curl China virus (TYLCCNV) and its associated betasatellite (TYLCCNB) attract their whitefly vectors by subverting plant MYC2-regulated terpenoid biosynthesis, therefore forming an indirect mutualism between virus and vector via plant. However, the evolutionary mechanism of interactions between begomoviruses and their whitefly vectors is still poorly understood. Here we present evidence to suggest that indirect mutualism may happen over a millennium ago and at present extensively prevails. Detailed bioinformatics and functional analysis identified the serine-33 as an evolutionary conserved phosphorylation site in 105 of 119 Betasatellite species-encoded βC1 proteins, which are responsible for suppressing plant terpenoid-based defense by interfering with MYC2 dimerization and are essential to promote whitefly performance. The substitution of serine-33 of βC1 proteins with either aspartate (phosphorylation mimic mutants) or cysteine, the amino acid in the non-functional sβC1 encoded by Siegesbeckia yellow vein betasatellite SiYVB) impaired the ability of βC1 functions on suppression of MYC2 dimerization, whitefly attraction and fitness. Moreover the gain of function mutation of cysteine-31 to serine in sβC1 protein of SiYVB restored these functions of βC1 protein. Thus, the dynamic phosphorylation of serine-33 in βC1 proteins helps the virus to evade host defense against insect vectors with an evolutionarily conserved manner. Our data provide a mechanistic explanation of how arboviruses evolutionarily modulate host defenses for rapid transmission.

## 1. Introduction

Arthropod-borne pathogens and parasites are major causal agents of diseases for humans, animals, and crop plants [1,2,3]. Of the 1600 species of known plant viruses, around 80% are arthropod-borne. The management of plant viral diseases is a global challenge. During infection, plant viruses induce several changes in the physiology of the host, which can influence the vector behavior and performance [4,5,6,7,8,9]. The relationships between viruses and their vectors can be mutualistic, neutral, or negative. Plant viruses can influence their vectors’ behaviors and fitness in two different ways. One is directly mediated by the virus in the vector’s body, for example, thrip-borne tospoviruses directly cause increased biting rates in infected vectors [10]. The other is indirectly mediated by altering plant traits as a consequence of virus infection. A mutualistic relationship may exist between some plant viruses and their vectors, which is mostly established via their shared plant hosts [2,11,12,13,14]. A few plant viruses could manipulate plant nutrition and volatile compound production to influence vector behaviors, including attractiveness, feeding behavior, and fecundity, to facilitate transmission [2,13,15]. Understanding the evolutionary mechanism of the indirect mutualism between some plant viruses and their vector insects is important for plant disease management since many invasive vector-borne viruses continue to emerge and spread between countries (e.g., whitefly transmitted begomoviruses [16,17]. Consequently, analysis of the evolutionary characteristics of various arboviruses plays an important role in exploring potential vector-borne virus prevention and control strategies.

The genus *Begomovirus* (family Geminiviridae) is a diverse group of circular single-stranded DNA viruses containing 445 species worldwide, according to the latest ICTV taxonomy [18]. Begomoviruses are exclusively transmitted through insect vector whitefly (*Bemisia tabaci*) persistently. Monopartite begomoviruses in the Old world possess only one component, DNA-A. Monopartite begomoviruses are often associated with symptom-determinant betasatellites. Betasatellites, belonging to the virus family *Tolecusatellitidae*, genus *Betasatellite*, generally enhance the accumulation of the helper virus and are required for symptoms [19,20]. There are currently 119 species in genus *Betasatellite*, and each betasatellite encodes a multifunctional βC1 protein. The βC1 proteins can suppress multiple layers of host antiviral defenses, including transcriptional gene silencing, post-transcriptional gene silencing, ubiquitination, autophagy, MAPK cascades, DNA methylation, and phytohormone signaling [2,21,22,23,24,25]. For example, we have identified several regulators (e.g., MYC2 and WRKY20) in host jasmonate (JA) signaling that have been widely hijacked by betasatellite-encoded βC1 and therefore establish an indirect mutualism with whitefly vector [4,13]. The evolutionary and biological significance of the other betasatellite species in a broad range in the tripartite interactions of virus-whitefly-plant is still elusive.

Viruses modulate the interactions between insect vectors and plants via the jasmonate, salicylic acid, and ethylene phytohormone pathways, resulting in changes in the fitness and viral transmission capacity of their insect vectors [2,16,26]. The JA signaling contributes as a key player in tackling both begomoviruses and whitefly vectors. This study explored whether other betasatellites modulate plant JA-regulated defense against whiteflies. Three subgroups of betasatellites are identified based on the amino acids sequence of βC1. We demonstrated that the betasatellite DNA-mediated suppression of JA signaling is common in the representative betasatellites of three subgroups, even in reputedly causing the earliest recorded plant virus disease over a millennium ago. The available phosphorylation state of βC1 at an evolutionarily conserved serine-33 is critical for JA signaling suppression. The mutations from serine-33 (S) to aspartate (D) or cysteine (C) in βC1 proteins weakened its ability to suppress olfactory defense and abolished its interference with MYC2 dimerization. Stable transgenic plant data also suggested that the dynamic phosphorylation of serine-33 of βC1 is essential for suppressing plant terpenoid-based defense. Our results provide a molecular target for controlling begomoviruses and managing whitefly, *B. tabaci* through semiochemicals.

## 2. Results

### 2.1. The Olfactory Attraction of the Whitefly Vector Is Conserved in All Betasatellite Subgroups

The betasatellites are essential pathogenicity determinants for begomoviruses with a typical feature of a single ORF βC1 encoded in the complementary-sense [27]. In the last five years, the number of betasatellites global isolates has increased from ~1300 to ~3100, and species have increased from 61 to 119. Field surveys have revealed that the outbreak of begomovirus-betasatellite diseases is often associated with the expansion of whitefly distribution [28,29,30]. Therefore, we speculate that the previous discovery of begomovirus-whitefly mutualism may be a conserved trait. Begomovirus-associated betasatellites are believed to have evolved independently and encode single conserved βC1 proteins [18,27,31], indicating that no suitable roots can be found in the phylogenetic tree for betasatellite species. Thus, to test this hypothesis, we performed a phylogenetic analysis of all available 119 βC1 proteins encoded by these distinct begomovirus-associated betasatellite species, using an unrooted tree. The phylogenetic analysis showed that betasatellite-encoded βC1 proteins could be divided into three subgroups (I, IIA, and IIB) (Figure 1A). Interestingly, the βC1 from a Japan-derived betasatellite (EpYVB, eβ) associated with a begomovirus Eupatorium yellow-vein virus (EpYVV), regarded as a causing pathogen of the earliest known record of a plant virus disease in 752 AD [32], and the recent China-derived Siegesbeckia yellow vein betasatellite (SiYVB, sβ) belonged to subgroup I. There was another African subclade in subgroup I of βC1 proteins, including African species such as cotton leaf curl Gezira betasatellite (CLCuGeB) and tomato leaf curl Ghana betasatellite 2 (ToLCuGHB2). Cotton leaf curl Multan betasatellite (CLCuMuB, cβ) with the most records in NCBI of 585 accessions, belonged to subgroup IIA. Compared to subgroups I and IIA, subgroup IIB contained more than half of the total numbers of betasatellite species, including our previously described Tomato yellow leaf curl China betasatellite (TYLCCNB, tβ).

One open question is how betasatellites and their encoded βC1 proteins have existed in nature and evolved with their hosts for such long time. To investigate whether functional evolution of betasatellites and βC1 proteins are related to the host preference of its whitefly vector, we performed insect preference assays for four betasatellites (including eβ, sβ, cβ, and tβ) belonging to three subgroups in *Nicotiana benthamiana* (Nb) plants. Given that betasatellites primarily rely on their helper viruses to spread, multiply, and encapsulate, and generally have a loose specificity by different helper components [33,34]. We chose Tomato yellow leaf curl China virus TYLCCNV DNA-A (TA) as a helper virus to explore the function of other betasatellites. As shown in Appendix A, plants inoculated with TA + tβ developed disease symptoms with severe curling of leaves and shot twisting, while only TA infection failed to cause symptoms. Plants inoculated with TA + cβ, TA + sβ, or TA + eβ displayed moderate leaf curling without shoot twisting. We further determined DNA accumulation of betasatellites in systemically infected leaves of plants infected with these four betasatellites in combination with the helper virus TA. The accumulation of tβ was higher in plants than in other betasatellites, which is consistent with the observed disease symptoms (Appendix A). Therefore, the helper virus TA can trans-replicate these four betasatellites in *Nicotiana benthamiana* (Nb), and some specificity exists for betasatellites trans-replication by begomoviruses. Similar to our previous observation on TA + tβ infection, TA + cβ complex-infected Nb attracted more whiteflies than vector control healthy Nb plants (Figure 1B). Intriguingly, two betasatellites from the subgroup I showed different attractions to the whitefly vectors. The TA + eβ complex-infected Nb could efficiently attract more whitefly, but the TA + sβ complex-infected Nb failed to attract more whitefly than those of vector control healthy Nb plants. Considering some specificity between betasatellites and the helper begomoviruses, we employed EpYVV DNA-A (EA) as the original helper virus of its associated betasatellite eβ to infect Nb plants. The eβ-mediated whitefly attraction was confirmed again. These results suggest that the begomovirus-whitefly mutualism might evolve a millennium ago when subgroup I betasatellites diverged.

Because plant terpenoid-related defenses play critical roles in the host preference, we performed volatile metabolite analysis to explore if the terpenoid-based defense still involves in the betasatellites-mediated whitefly attraction. A pivotal plant volatile sesquiterpene α-bergamotene, its emission significantly decreased on TA + tβ, TA + cβ, TA + eβ, and EA + eβ-infected Nb than that of vector control healthy Nb plants (Figure 1C,D). Consistent with TA + sβ infection displayed no changes in whitefly attraction, the level of α-bergamotene emission in TA + sβ-infected Nb plants showed similar to that of vector control healthy Nb plants. As expected, the transcription levels of sesquiterpene synthase gene *NbTPS1* were significantly reduced in plants infected by begomovirus-betasatellite complexes, except for TA + sβ infection, which did not affect *NbTPS1* expression (Appendix A). To further explore whether the attraction by betasatellites is specific to the whitefly vector, we performed insect choice assays with begomovirus non-vector green peach aphid (*Myzus persicae*). More non-vector aphids were preferred on uninfected plants compared to TA + tβ infected plants (Appendix A). These results are consistent with the fact that α-bergamotene has been reported as an attractant to aphids [35]. These results suggest that whitefly attraction is specifically conserved in betasatellites from all three subgroups I, IIA, and IIB, by regulating plant terpenoid-based defense.

### 2.2. The Evolutionarily Conserved Serine-33 Is Essential for βC1-Mediated Whitefly Attraction

The above results show that siegesbeckia yellow vein betasatellite sβ belonging to betasatellite subgroups I do not possess traits of whitefly attraction caused our interest in understanding the behind mechanism. Mutations are considered one crucial factor in genetic diversity among plant viruses [17]. All functions of betasatellites are accredited to a single complementary protein βC1, and the phosphorylation modification on βC1 protein is a major antiviral plant response and pathogenesis [36,37,38]. We thus wanted to determine if some mutations on phosphorylation sites of βC1 protein make sβ not change the host preference. To test this, we analyzed the primary amino acid sequence of βC1 proteins encoded by 119 betasatellite molecules and predicted potential phosphorylation sites using the NetPhos 3.1 server (http://www.cbs.dtu.dk/services/NetPhos; accessed on 22 September 2021). Each βC1 protein has several evolutionarily conserved phosphorylation sites, among which serine-33 is the most conserved residue in 105 of 119 *Betasatellite* species in the three subgroups (Appendix A). Intriguingly, most of βC1 proteins with non-Serine-33 in 14 *Betasatellite* species were found in subgroup I, esp. the African subclade and sβ isolated from Asia. The phosphorylation mimics impair the functions of βC1 as a viral suppressor of RNA silencing and a symptom determinant [36,37]. To identify whether the serine-33 of βC1 functions has a critical role in the olfactory attraction of whitefly, we mutated the serine-33 to aspartate (S33D) to mimic the phosphorylation of βC1 proteins in tβ and eβ and named as tβ^S33D^ and eβ^S33D^. When the begomovirus-betasatellite complexes TA + tβ^S33D^ and EA + eβ^S33D^ infected Nb plants, we observed that TA + tβ^S33D^ and EA + eβ^S33D^ abolished betasatellite-mediated whitefly vector attraction (Figure 1B). The emission of α-bergamotene and the expression of *NbTPS1* on TA + tβ^S33D^ and EA + eβ^S33D^ infected Nb was also comparable to healthy plants, which was significantly higher than that of TA + tβ and EA + eβ-infected Nb plants (Figure 1D and Appendix A). These results suggest that an evolutionarily conserved phosphorylation site serine-33 of βC1 protein is essential for betasatellite-mediated whitefly attraction.

To further confirm the conserved phosphorylation site serine-33 of βC1 protein-mediated vector choice, we used the heterologous Potato virus X (PVX) model system for systemic ectopic expression of betasatellited-encoded βC1 proteins and βC1 mutations through agro-inoculation of Nb plants. Plants inoculated with PVX-GFP served as a control. There were no obvious morphological differences between these recombinant PVX vector-infected plants (Appendix A). We performed the two-choice assay to determine whether the expression of a single βC1 protein is sufficient to affect the whitefly attraction. Similar to the whitefly attraction toward the host infected with begomovirus-betasatellite complex, more whiteflies were attracted to Nb plants infected with recombinant PVX-eβC1, PVX-tβC1 compared to PVX-GFP control plants (Figure 2A). Whereas PVX-eβC1^S33D^ and PVX-tβC1^S33D^-infected Nb plants significantly decreased whitefly attraction compared with PVX-eβC1 and PVX-tβC1-infected Nb plants. Consistent with what we have observed on the begomovirus TA + sβ complex-infected plants, PVX-sβC1-infected plants could not change the whitefly preference. If the βC1 proteins with serine-33 available for phosphorylation are vital for whitefly attraction, we considered that the substitution of serine-33 with cysteine-31 in the sβC1 protein could compromise the ability to attract whitefly and suppress plant terpene-related defense. Thus, we postulated that mutation of cysteine-31 to a serine residue in sβC1 protein (sβC^C31S^) could restore its function on whitefly attraction. Indeed, we observed that PVX-sβC1^C31S^-infected plants attracted more whiteflies than those of PVX-sβC1 and PVX-GFP-infected plants (Figure 2A). Consistent with the effect on changes in the whitefly olfactory behavior, changes in α-bergamotene emission and *NbTPS1* expression in infected plants are relevant to the specific variant of serine-33 in βC1 proteins. Similar to PVX-eβC1 and PVX-tβC1 infection, PVX-sβC^C31S^ infection displayed a decrease of α-bergamotene emission and downregulation of *NbTPS1* expression compared to PVX-GFP infection (Figure 2B–D). PVX-eβC1^S33D^ and PVX-tβC1^S33D^ infection exhibited much fewer effects on α-bergamotene emission and *NbTPS1* expression than those of PVX-GFP infected plants. These results further demonstrate that the dynamic phosphorylation state of serine-33 in βC1 proteins confers the evolutionarily conserved whitefly attraction by the begomovirus-betasatellite complex.

### 2.3. The Conserved Serine-33 in βC1 Proteins Is Responsible for Suppressing MYC2 Dimerization

We previously demonstrated that βC1 interacts with MYC2 to suppress plant terpenoid-based defense and promotes whitefly preference [13]. The phosphorylation mimics of βC1 abolish its interaction with ASYMMETRIC LEAVES 1 (AS1) to attenuate virus disease symptoms in *N. benthamiana* leaves [37]. We asked whether the phosphorylation mimics of βC1 proteins at the conserved serine-33 also alter its interaction with MYC2 to affect plant terpenoid-based defense and whitefly preference. To investigate this, we determined the interaction of tβC1 and its phosphorylation mimics with AtMYC2 by a bimolecular fluorescence complementation (BiFC) assay. *Agrobacterium tumefaciens* strains containing expression vectors for tβC1-cEYFP or tβC1^S33D^-cEYFP with nEYFP-MYC2 were co-infiltrated into *N. benthamiana* leaf cells. The strong interaction (represented by fluorescence) between tβC1-cEYFP and nEYFP-MYC2 or tβC1^S33D^-cEYFP and nEYFP-MYC2 was observed in nuclei (Appendix A), which has been confirmed by our previous study with DAPI staining [13]. As a negative control, no fluorescence was observed when tβC1-cEYFP or tβC1^S33D^-cEYFP was coexpressed with the nEYFP-GUS vector. This conserved serine-33 residue of βC1 proteins does not play an essential role in its interaction with MYC2.

MYC2 dimerizations contribute to the binding to the G-boxes in the promoter of its downstream target genes and regulate multiple signaling events [39,40]. Given that the TYLCCNB-encoded tβC1 suppresses MYC2 activity by interfering with MYC2 dimerization [13], we hope that the conserved serine-33 in βC1 proteins are responsible for interfering with MYC2 dimerization. To examine this hypothesis, we performed a modified BiFC assay following our previous report [4]. In the presence of coexpressing tβC1, cβC1, and eβC1, the interaction signal strength of MYC2-MYC2 decreased to approximately half of its original intensity (Figure 3A,B). These results confirmed the conserved function of betasatellite-encoded βC1 proteins on the interference with MYC2 dimerization.

Additionally, the phosphorylation mimics had a minor influence on the suppression of MYC2 dimerization compared with its wild-type βC1 proteins. sβC1 did not affect the formation of MYC2 dimerization, but the gain of function sβC1^C31S^ harbored a similar effect on suppression of MYC2 dimerization to the other three βC1 proteins. Furthermore, an in vitro competitive pull-down assay further confirmed the above results. The amount of GST-MYC2 pulled down by MBP-MYC2 was reduced by increasing the amount of His-tβC1 in the mix reaction (Figure 3C). By contrast, increasing the amount of His-tβC1^S33D^ protein showed less effect on MYC2 self-association than adding the same content of His-tβC1 in the reaction. Together, these results suggest that the conserved serine-33 is a key residue in βC1 proteins for interfering with MYC2 dimerization.

Next, we asked whether the conserved residue of βC1 proteins at serine-33 influences the MYC2 trans-activation activity. Using *TPS10 promoter*: *Luciferase* (LUC) as a reporter, and MYC2, tβC1, and tβC1^S33D^ as effectors, we transiently expressed with the indicated effector and reporter constructs in *N. benthamiana* leaf cells. GUS was used as the control effector. LUC activities were subsequently quantified. tβC1coexpressing with MYC2 significantly decreased the LUC activity, whereas tβC1^S33D^ coexpression did not show significant differences in MYC2-induced LUC activity compared to GUS coexpression (Figure 3D,E). These results indicate that the conserved serine-33 also plays a key role in βC1 suppression of MYC2 trans-activity.

### 2.4. The Conserved Serine-33 in βC1 Proteins Contributes to Its Compacity to Affect Whitefly Performance

To further confirm the importance of serine-33 sites of βC1 on whitefly performance, we conducted whitefly two-choice assays and bioassays using stable βC1-transgenic *Arabidopsis thaliana.* Consistent with these above results on virus-infected *N. benthamiana* (Figure 1B and Figure 2A), eβC1 transgenic *Arabidopsis* was more attractive to whitefly compared to wild type Col-0, while eβC1^S33D^ transgenic *Arabidopsis* attracted fewer whiteflies compared to Col-0 and eβC1 (Figure 4A), indicating that the phosphorylation of βC1 at serine-33 disrupts βC1-mediated whitefly preference. We note that sβC1 transgenic *Arabidopsis* failed to attract more whiteflies, but sβC1^C31S^ transgenic *Arabidopsis* enhanced whitefly attraction compared to Col-0 and sβC1. These results indicate that the dynamic phosphorylation of serine-33 in βC1 proteins commonly renders the host more vector attraction through the evolution of betasatellites in nature.

Because βC1 suppresses the activity of MYC2, which is involved in regulating multiple secondary metabolites against herbivore insects [40], to investigate whether the conserved serine-33 of βC1 is involved in plant resistance to whiteflies, we performed whitefly bioassays using Col-0 and βC1-overexpressing *Arabidopsis* plants. Whiteflies laid more eggs on tβC1, eβC1, and sβC1^C31S^ transgenic plants than on Col-0 (Figure 4B). Conversely, similar eggs were laid by whiteflies on tβC1^S33D^, eβC1^S33D^, and sβC1 transgenic plants compared to on Col-0 plants. We thus considered that whitefly development could be affected by expressing βC1 proteins in plants. As expected, whitefly developed significantly faster on tβC1, eβC1, and sβC1^C31S^ transgenic plants than on Col-0 plants, as indicated by more pupae (Figure 4C). Whitefly development on tβC1^S33D^, eβC1^S33D^, and sβC1 transgenic plants progressed similarly to Col-0. These data suggest that the dynamic phosphorylation of serine-33 in βC1 proteins contributes to whitefly performance in plants.

## 3. Discussion

Emerging and re-emerging insect-borne viral diseases severely burden global public health and agricultural production. Understanding virus–host–vector complex interactions, in particular, the effects of viral infection on host–insect interactions, would be greatly helpful in designing antiviral strategies [2,16,41]. It seems that the emergence of begomovirus-betasatellite complexes during the past years is closely associated with the prevalence of the whitefly vector, esp. the invasive B and Q biotypes [16,42,43]. We have previously shown interactions between βC1 and several plant proteins, including three transcription factors (MYC2, WRKY20, and PIFs) [4,13,44]. Those hijacking of host defense systems would benefit insect vectors in many ways: attract insect vector whitefly, deter non-vector, and environmental adaptation (esp. light conditions). Here we demonstrated that the dynamic phosphorylation of serine-33 in βC1 proteins is important for interfering with the transcription activation of MYC2-regulated terpenoid synthesis. Mechanistically, the phosphorylation state of serine-33 coupled with selection pressure may play important roles in shaping the co-evolutionary outcome between begomoviruses and whitefly.

It is interesting to identify further the relationship between the biochemical activity of βC1 and its known functions in RNA silencing suppression and interaction with host autophagy-related immunity. The phosphorylation mimics weakened the ability of TYLCCNB-encoded tβC1 to suppress gene silencing and abolished its interaction with a leaf stem cell regulator-ASYMMETRIC LEAVES 1 (AS1) in *N. benthamiana* leaves [37]. AS1 negatively regulates the expression of *PDF1.2* and *PR4* in JA signaling. Our previous studies also show that the βC1 interaction partners AS1 and MYC2 mediate different JA signaling branches against whitefly [13]. The two α-helix structures of βC1 (60aa–100aa) contribute to the heterodimerization with MYC2. In this study, we found that both the N-terminal and the C-terminal of βC1 contribute to the dimerization with MYC2. Especially, the dynamic phosphorylation of serine-33 in βC1 proteins strongly affects the dimerization of MYC2 and the virus-vector mutualism. Interestingly, a mutation near serine-33 in CLCuMuB-encoded βC1 protein (cβC1^V32A^) abolished its interaction with NbATG8 and the JA responses [45]. Meanwhile, the mimic phosphorylation of TYLCCNB-βC1 had no impact on its protein stability, subcellular localization, or self-association [37]. Therefore, the phosphorylation modifications on viral proteins may also affect βC1 interaction with autophagy and JA signaling pathway.

Post-translational modifications, especially phosphorylation, regulate the “arms race” between viruses and plants/animals [46]. The emergence of cooperation becomes interesting when virus–vector interactions are based on coordinative defense against their shared host plant. Plant terpenoids have been identified to be involved in the whitefly attraction, and terpenes’ emission requires induction by some chemicals such as MeJA [2,16]. Our previous research showed that MYC2 mediates terpenoid biosynthesis by direct regulating *TPS* genes to defend against whitefly host preference [13]. In this study, we performed a luciferase assay to examine if the phosphorylation of MYC2 affects its-regulated terpene defense. The phosphorylation levels of MYC2 are relative to its transcription activation, as indicated that phosphorylation mimic MYC2^S3D^ promoted its trans-activity in the *TPS10* promoter, and the low phosphorylation mutant MYC2^S3A^ reduced this trans-activity compared to wild-type MYC2 (Appendix A). These results strongly suggest a tight relationship between the phosphorylation and the tripartite interaction of the plant–virus–insect vector.

Given the strong whitefly vector restriction of the begomovirus-betasatellite complex, it is likely that the primary evolutionary imperative of betasatellites is the maintenance of optimal virus–vector interactions, which is similar to most plant viruses that are persistently transmitted by one or a small number of vectors [17]. All betasatellites encode βC1 protein on the complementary strand. Recent studies describe that approximately 40% of reported betasatellite (e.g., TYLCCNB) sequences encode a βV1 protein on the viral strand, which also contributes to symptom development [47]. We notice that most of betasatellites in Subgroup IIB can encode βV1 protein, but many betasatellites belonging to the other two subgroups cannot encode βV1 protein, including EpYVB, SiYVB, and CLCuMuB. Further studies need to explore whether βV1 plays a role in virus–plant–vector interactions. We have proved that the substitution residues at serine-33 with either aspartate or cysteine influence the dynamic phosphorylation of βC1 proteins, thereby damaging virus–vector mutualism. Thus, betasatellites without serine-33 including SiYVB, are limited in nature, which might be due to a weaker competitive advantage for parasite transmission. These betasatellites without serine-33 mainly localize in southwest Asia and west Africa (Appendix A), which is consistent with the invasion route of begomoviruses and whitefly vector, indicating the evolutionary direction of betasatellites is the maintenance of the cooperation between begomoviruses and whitefly vector. The current study implies that whitefly-transmitted begomovirus disease complexes were prevalent before modern intensive agricultural practices, which have encouraged the spread and diversification of geminiviral diseases.

## 4. Materials and Methods

### 4.1. Plant, Virus, and Whitefly Materials and Growth

*Nicotiana benthamiana* plants were grown under controlled conditions at 25 °C in a growth chamber with 14 h dark/10 h light photoperiod. *Arabidopsis thaliana* plants were grown at 22 °C in a growth chamber with 14 h dark/10 h light photoperiod. Several betasatellite-encoded βC1 and its mutants transgenic *A. thaliana* plants were performed using the floral-dip method. Infectious clones of TYLCCNV (AJ319675.1) and TYLCCNB (AJ781200) were described previously [48]. Infectious clones of EpYVV (AB007990.1), EpYVB (AJ438938.1), SiYVV (NC_038682.1), and SiYVB (JF682839.1) and all recombinant viruses with point-mutation were constructed and generated in this report. MEAM1 (*Middle East-Asia Minor 1*) whiteflies were collected from the suburbs of Beijing, China. The whitefly population was maintained on healthy cotton plants grown in a growth chamber (26 °C, 50% RH) with a 12 h light/12 darkness cycle.

### 4.2. Virus Inoculation

*Agrobacterium tumefaciens* strain C58C1 was transformed with different constructs using electroporation. *A. tumefaciens* cultures for inoculation were grown at 28 °C for 48 h to an optical density of 1.0–1.5 at 600 nm with appropriate antibiotic concentrations. The bacterial cells were pelleted (5000× *g* for 15 min at 20 °C) and resuspended in *MMA buffer* (10 mM MES, 10 mM MgCl_2_, 200 μM AS). Cells were incubated in MMA medium for 3 h and then infiltrated into young and fully expanded leaves of 3 to 4 weeks old *N. benthamiana* plants using 1 mL syringe. For begomovirus-betasatellite complex infection experiments, *N. benthamiana* leaves were inoculated with *Agrobacterium* EHA105 carrying TYLCCNV (TA) and TYLCCNB (TA + tβ), with the phosphorylation mimic mutant tβC1 betasatellite (TA + tβ^S33D^), with CLCuMuB (TA + cβ), with EpYVB (TA + eβ), or with SiYVB (TA + sβ), with gain-of-function mutant (TA + sβ^C31S^). *N. benthamiana* leaves were also inoculated with *Agrobacterium* EHA105 carrying EpYVV DNA-A (EA) with EpYVB (EA + eβ), with the phosphorylation mimic mutant eβC1 betasatellite (EA + eβ^S33D^). Total genomic DNA was extracted from systemically infected leaves by CTAB. The viral DNA was detected by real-time PCR with TYLCCNV-specific primers. *N. benthamiana EF1a* was used as an endogenous control.

### 4.3. Phylogenetic Analysis

Around 3100 betasatellite sequences belonging to 119 species of distinct betasatellites are present in the NCBI database. The amino acid sequences of 119 betasatellite species were downloaded from the NCBI database and βC1 sequences were isolated. The evolutionary history was inferred using the Maximum Parsimony method. The consistency index is 0.379195, the retention index is 0.717269, and the composite index is 0.275199 (0.271985) for all sites and parsimony-informative sites (in parentheses). The percentage of replicate trees in which the associated taxa clustered together in the bootstrap test (1000 replicates) are shown next to the branches. The MP tree was obtained using the Close-Neighbor-Interchange algorithm with search level 3 in which the initial trees were obtained with the random addition of sequences (10 replicates). All positions containing gaps and missing data were eliminated from the dataset (Complete Deletion option). There is a total of 88 positions in the final dataset, out of which 83 were parsimony informative. Phylogenetic analyses were conducted in MEGA4 with 1000 bootstrap values through the method based on betasatellite-encoded βC1 protein. The scale bar represents 0.2 amino acid substitutions per site in the primary structure.

### 4.4. Insect Preference Experiments

The whitefly two-choice experiments were performed as described previously [13]. Two plants with similar sizes and leaf numbers were placed in an insect cage (30 × 30 × 30 cm). Two hundred adult whiteflies were released in the middle of the two plants. After 25 min, the number of whiteflies settled on each plant was recorded. Six biological replicates were conducted in this experiment. A short-term two-choice experiment using green peach aphids was performed as described previously [4]. Two plants of similar size and leaf number were placed in an insect cage (30 × 30 × 30 cm), and 50 adult green peach aphids were released into the cage. Twenty-four hours after release, the number of aphids settled on each plant was recorded. Six biological replicates were conducted in this experiment.

### 4.5. Volatile Enalysis

Isolation, identification, and analysis of volatile compounds from *N. benthamiana* plants were performed as described previously [13]. Volatiles emitted from individual plants infected by begomovirus-betasatellite complex or recombinant Potato virus X (PVX) were collected for 12 h with a solid phase microextraction fiber (SPME; Supelco, Belafonte, PA, USA) consisting of 50/30 μm DVB/CAR/PDMS (Supelco, 57328-U, USA). Plant volatiles were analyzed by gas chromatography-mass spectrometry (GC-MS) (Shimadzu, QP2010, Japan) coupled with a DB5MS column (Agilent, Santa Clara, CA, USA, 30 m × 0.25 mm × 0.25 µm). Oven temperature in gas chromatography was as follows: 40 °C for 3 min, raised at 5 °C/min to 250 °C, and then held for 5 min. Four plants of each infection were used.

### 4.6. Quantitative PCR

Total RNA was isolated from *N. benthamiana* plants using the RNeasy plant mini kit (Qiagen, 74904, Germany) including on-column DNase treatment. Reverse transcription was performed using 2 μg RNA of each sample and oligo (dT)_18_ primers by M-MLV reverse transcriptase (Promega, 28025013, USA). Three independent biological samples were collected and analyzed. Real-time PCR was performed on a Bio-Rad CFX96 real-time PCR system with Thunderbird^TM^ SYBR qPCR Mix (TOYOBO, QPS-201, Japan) and gene-specific primers listed in Appendix A. The *N. benthamiana EF1a* was used as an internal control.

### 4.7. Plasmid Construction and Agroinfiltration

For the construction of the recombinant PVX virus expression vectors, the βC1 and its mutant genes were cloned into the PVX vector pGR208 using gene-specific primers provided in Appendix A. For transient expression analysis, DNA fragments were PCR amplified and cloned into a pENTR-3C entry vector. These fragments were then shifted to a binary vector under the control of a CaMV 35S promoter. All constructs used for protein expression in plants were transformed into *Agrobacterium tumefaciens* strain EHA105. *Nicotiana benthamiana (Nb)* leaves were inoculated with recombinant Potato virus X (PVX) vectors PVX-eβC1, PVX-eβC1^S31D^, PVX-sβC1, PVX-sβC1^C31S^, PVX-tβC1, PVX-tβC1^S33D^ via agroinfiltration. PVX-GFP was used as the control.

### 4.8. Preparation of Recombinant Proteins

DNA fragments encoding full-length tβC1 were amplified by PCR using Phusion high-fidelity DNA polymerase (Thermo Scientific, F530S, USA) using primers listed in Appendix A, and amplified DNA fragment was ligated into pET-28a (+) to generate GST fusion or His fusion constructs. A point mutation in specific amino acids of tβC1^S33D^ was generated by a site-directed mutagenesis kit (Sangon Biotech, Shanghai, China). All constructs were transformed into *Escherichia coli* BL21 (DE3) competent cells, and recombinant proteins were purified from bacterial extracts after isopropyl-β-D-thiogalactoside (IPTG) induction. Recombinant GST tag proteins were purified using glutathione sepharose (GE Healthcare, 17-0756-01, USA) beads according to the manufacturer’s instructions. His tag proteins were purified using Ni-nitrilotriacetate (Ni-NTA) agarose (Qiagen, 30210, Germany) according to the manufacturer’s instructions.

### 4.9. In Vitro Pull-Down Assay

The in vitro binding assay was performed as previously described [13]. Pull-down proteins were separated on 10% SDS-polyacrylamide gels and detected through immunoblotting with anti-GST antibody (TransGen Biotech, HT601-02, Beijing, China).

### 4.10. Bimolecular Fluorescence Complementation (BiFC) Assay

Amplified fragments of MYC2 were cloned into vectors to generate fusion genes with nEYFP or cEYFP at the amino- or carboxy-terminus with a cauliflower mosaic virus 35S promoter. The tβC1 and tβC1^S33D^ were cloned into vectors to generate fusion genes with cEYFP. All constructs were transferred into *Agrobacterium* C58C1 competent cells. The BiFC assay was performed as previously described [13]. *Agrobacterium* cells containing indicated constructs were infiltrated into the leaves of three-week-old *N. benthamiana*. Images of fluorescence were taken using confocal microscopy (Leica SP8) after 48 h incubation.

### 4.11. Luciferase Assays

*N. benthamiana* leaves were agroinfiltrated with *A. tumefaciens* EHA105 strains carrying different combinations of DNA constructs as indicated in the Table/Figures. Forty-eight hours after infiltration, leaves of Nb expressing different constructs were harvested and assayed for luciferase (LUC) activity with eight independent biological replicates. *TPS10* promoter: LUC was used as a reporter construct. The CaMV 35S promoter-driven MYC2 and its mutants or tβC1 and tβC1^S33D^ were used as effector constructs, and CaMV 35S promoter-driven GUS was used as the control.

### 4.12. Whitefly Bioassays

We performed the whitefly oviposition and development experiments according to the previous method using plant leaves of similar size and a leaf-clip cage [13,48]. For the whitefly oviposition experiment, three male and female adult whiteflies were released into a leaf-clip cage. Whitefly eggs on the leaf area enclosed in a cage were counted after 10 days. For whitefly development experiment, 16 adult female whiteflies were released into a leaf-clip cage and then removed after 2 days of oviposition. The numbers of whitefly pupae developed from the eggs oviposited in each of the leaf cages were recorded after 20 days.

### 4.13. Data Analysis

Differences in insect two-choice between different lines were analyzed by Wilcoxon matched pairs tests. Differences in gene expression level and relative virus titer were analyzed using two-tailed Student’s *t*-tests for comparing two treatments or two lines. Significant differences in gene expression level, volatile analysis, number of MYC2 dimerization, *TPS10* promoter: *luciferase* activity, number of whitefly oviposition and development among different treatments or lines were tested using one-way ANOVA followed by Duncan’s multiple range tests.

## Figures and Tables

**Figure 1 cells-12-00149-f001:**
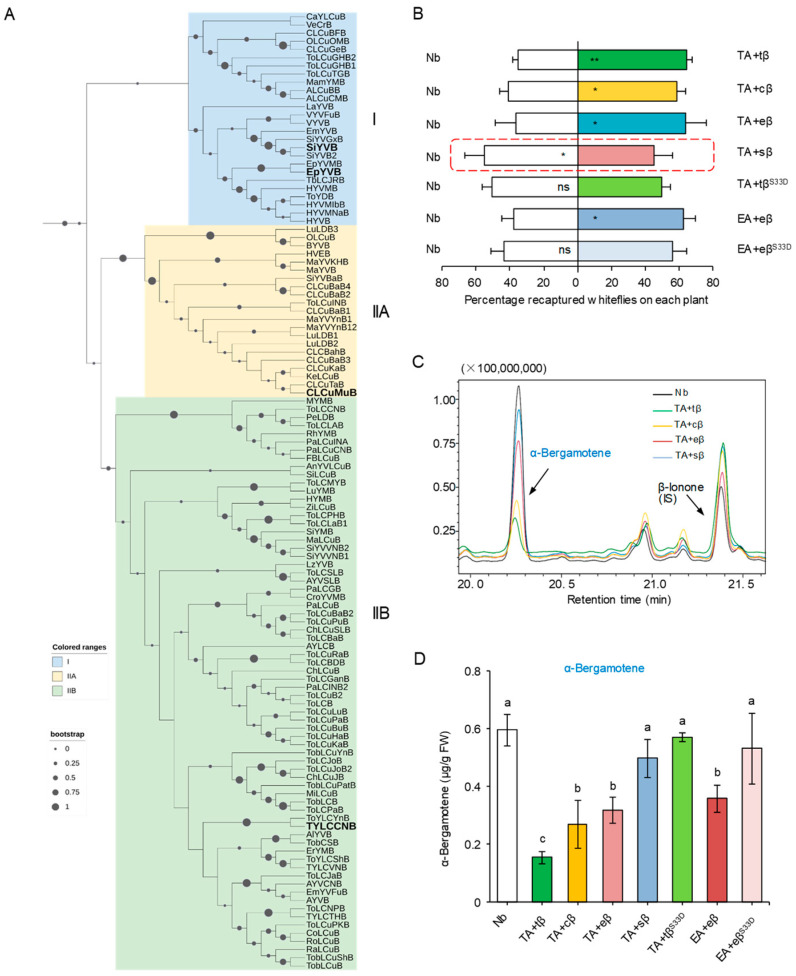
The evolutionarily conserved attraction of whitefly by begomovirus-associated betasatellites. (**A**) Neighbor-joining phylogenetic tree of 119 betasatellite-encoded βC1 proteins was constructed using MEGA4 based on multiple protein sequence alignments made with ClustalX. Selected protein accession numbers are according to the ICTV Report in Appendix A. (**B**) Whitefly preference (as percentage recaptured whiteflies out of 200 released) on each plant. Uninfected *N. benthamiana* (Nb) and begomovirus-betasatellite complex-infected plants were used for two-choice experiments for whitefly. Selected viruses were recorded as follows: TYLCCNV (TA), TYLCCNB (tβ), CLCuMuB (cβ), EpYVB (eβ), SiYVB (sβ), EpYVV/EpYVB complex (EA + eβ). Phosphorylation mimic βC1 mutants were named as TYLCCNB βC1^S33D^ (tβ^S33D^), EpYVB βC1^S31D^ (eβ^S31D^). Values are means+ SE (*n* = 6) (ns, no significant differences; * *p* < 0.05, ** *p* < 0.01; Wilcoxon matched pairs test). (**C**) Typical gas chromatograms (GC) of terpenes emitted by uninfected-*N. benthamiana* (Nb) and begomovirus-betasatellite complex-infected Nb plants. β-Ionone was used as an internal standard (IS). (**D**) α-Bergamotene emitted by Nb plants or begomovirus-betasatellite complex-infected plants. Bars represent means ± SD (*n* = 4). Lowercase letters indicate significant differences among columns according to one-way ANOVA followed by Duncan’s multiple range test (*p* < 0.05).

**Figure 2 cells-12-00149-f002:**
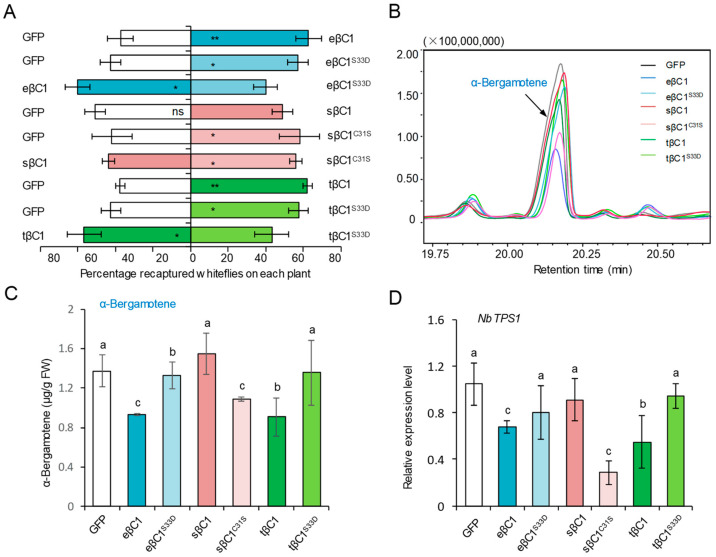
A conserved serine-33 site in βC1 proteins is crucial for the evolutionary betasatellite-mediated plant attraction to whitefly vector. (**A**) The preference of whitefly vector on *N. benthamiana* inoculated with *Agrobacteria* carrying individual recombinant PVX vectors at 10 dpi. *Nicotiana benthamiana* (*Nb*) leaves were inoculated with recombinant Potato virus X (PVX) vectors PVX-eβC1, PVX-eβC1^S31D^, PVX-sβC1, PVX-sβC1^C31S^, PVX-tβC1, PVX-tβC1^S33D^ via agroinfiltration. PVX-GFP was used as the control. Bars represent means ± SD (*n* = 6) (ns, no significant differences; * *p* < 0.05, ** *p* < 0.01; Wilcoxon matched pairs test). (**B**) Chromatograms display differences in indicated volatile compounds between PVX-GFP- and PVX-βC1-infected plants. β-Ionone was used as an internal standard (IS). (**C**) α-Bergamotene emitted by recombinant PVX-infected plants. Bars represent means ± SD (*n* = 4). Lowercase letters indicate significant differences among columns according to one-way ANOVA followed by Duncan’s multiple range test (*p* < 0.05). (**D**) Relative expression level of *NbTPS1* in recombinant PVX-infected plants. Bars represent means ± SD (*n* = 4). Lowercase letters indicate significant differences among columns according to one-way ANOVA followed by Duncan’s multiple range test (*p* < 0.05).

**Figure 3 cells-12-00149-f003:**
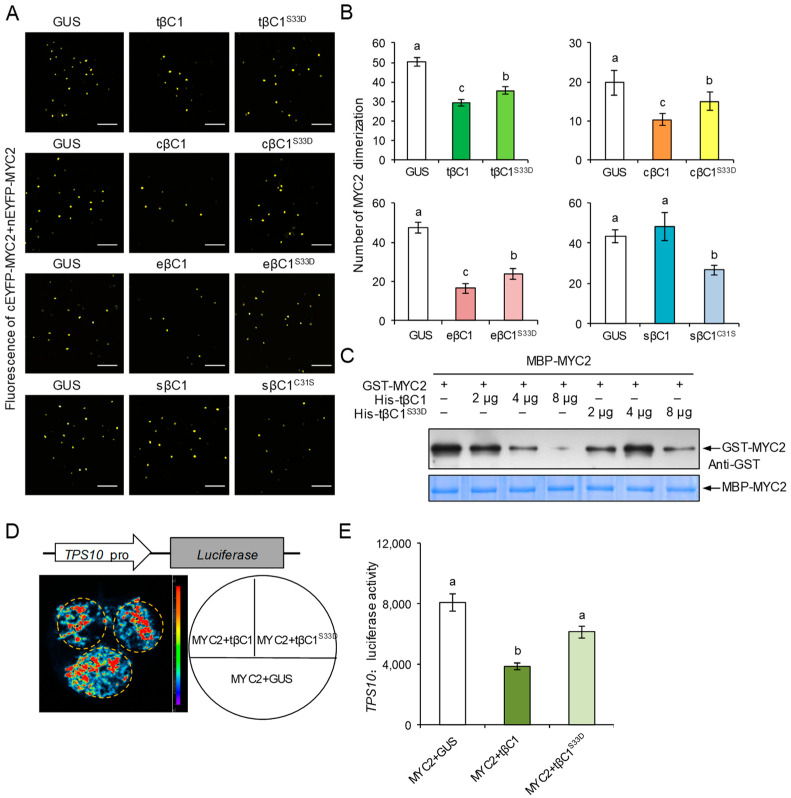
The conserved serine-33 site of the βC1 protein interferes with the transcription activation of MYC2-regulated terpenoid synthesis genes. (**A**) Effects of serine-33 in βC1 on the suppression of MYC2 dimerization by BiFC competition assays. The EYFP fluorescence was detected after co-expression of GUS, different betasatellite-encoded βC1, and its mutants with cEYFP-AtMYC2 and nEYFP-AtMYC2. Scale bars, 100 μm. (**B**) Effects of serine-33 in βC1 on the dimerization of MYC2. Bars represent means ± SD (n = 8). Lowercase letters indicate significant differences among columns according to one-way ANOVA followed by Duncan’s multiple range test (*p* < 0.05). (**C**) Pull-down protein competition assays. The indicated protein amount of His-tβC1 or His-tβC1^S33D^ was mixed with GST-MYC2 and pulled down by MBP-MYC2. Immunoblots were performed using anti-GST antibodies to detect the associated proteins. Membranes were stained with Coomassie Brilliant Blue to monitor the input protein amount. (**D**) Luciferase imaging of phosphorylation mimic mutant tβC1^S33D^ influencing MYC2 transcriptional activity on *TPS10* promoter in *N. benthamiana*. *N. benthamiana* leaves infiltrated *TPS10* promoter: *LUC* and *35S: MYC2* together with *35S: tβC1*, *35S: tβ^S33D^*, or *35S: GUS* and were subjected to luciferase complementation imaging assay. (**E**) The effect of tβC1 and tβC1^S33D^ on the transcriptional activity of MYC2 on *TPS10* promoter. *N. benthamiana* leaves infiltrated with indicated constructs were sliced into strips, and relative luminescence was determined by a microplate luminometer. Values are mean ± SEM (n = 8). Lowercase letters indicate significant differences among columns according to one-way ANOVA followed by Duncan’s multiple range test (*p* < 0.05).

**Figure 4 cells-12-00149-f004:**
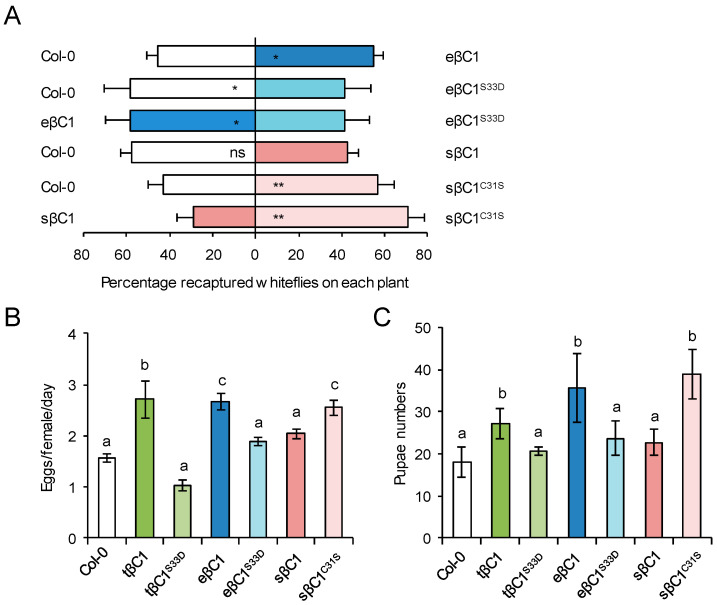
A conserved serine-33 site in βC1 proteins is crucial for maintaining virus-vector coevolved adaptations. (**A**) The preference of whitefly vector on *Arabidopsis* transgenic expressing different betasatellite-encoded βC1 and its mutants (eβC1, eβC1^S31D^, sβC1, sβC1^C31S^). Col-0 was used as the control. Bars represent means ± SD (n = 6) (ns, no significant differences; * *p* < 0.05, ** *p* < 0.01; Wilcoxon matched pairs test). (**B**) Number of eggs laid per female whitefly per day on Col-0, different betasatellite-encoded βC1 and its mutants expressing transgenic *Arabidopsis* plants. (**C**) Pupae numbers of whiteflies on Col-0, different betasatellite-encoded βC1 and its mutants expressing transgenic *Arabidopsis* plants. In (**B**,**C**), bars represent means ± SD (n = 6). Lowercase letters indicate significant differences among columns according to one-way ANOVA followed by Duncan’s multiple range test (*p* < 0.05).

## Data Availability

The data presented in this study are available on request from the corresponding author.

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
