# Peer review of "Diverse Begomoviruses Evolutionarily Hijack Plant Terpenoid-Based Defense to Promote Whitefly Performance"

_cells, 2022, doi:10.3390/cells12010149_

Round 1

Reviewer 1 Report

In this study, authors identified the serine-33 as an evolutionarily conserved phosphorylation site for most Betasatellite species-encoded BC1 proteins that inhibit plant terpenoid-based defense by interfering with MYC2 dimerization and are essential to promote whitefly performance. The substitution of serine-33 of BC1 proteins affected the BC1 functions on suppression of MYC2  dimerization, whitefly attraction, and fitness. The dynamic phosphorylation of serine-33 in βC1 proteins ensures the evolutionarily indirect mutualism between the begomovirus-betasatellite complex and whitefly vector. This study reveals that arboviruses evolutionarily modulate host defenses for rapid transmission. Generally speaking, it is a very nice manuscript. I have a few comments for authors before publication in this journal.

Fig 3A, the confocal images need to be clarified; please provide high-quality images.

Line 104, Please explain that the nuclear shuttle protein (BV1) encoded by bipartite begomovirus DNA-B of the SLMV was used as a root for constructing the phylogenetic tree.

Line 110: ‘Asia-subclade.’?  . is necessary.

Line 739, ‘which lost changes into ‘without’

The formats of references could be more consistent. Please double-check all of them. For example, Trends in plant science, PloS one, etc.. are not standard abbreviations for these journals.

Author Response

Q1: Fig 3A, the confocal images need to be clarified; please provide high-quality images.

Response: Thanks for your suggestion. Here we have replaced these images with high-quality ones in the new version manuscript.

Q2: Line 104, Please explain that the nuclear shuttle protein (BV1) encoded by bipartite begomovirus DNA-B of the SLMV was used as a root for constructing the phylogenetic tree.

Response: Thank you for your critical comments. In the revision, we have clarified the reasons for SLCMV BV1 as an outgroup (not a root) in the phylogenetic tree.

(1) Begomoviruses are widely distributed and causing devastating diseases in many crops. According to the number of genomic components, a begomovirus is known as either monopartite or bipartite begomovirus. Both the monopartite and bipartite begomoviruses have the DNA-A component. Bipartite begomoviruses contain the DNA-B component. Monopartite begomoviruses are associated with a group of ssDNA satellites almost half of their helper begomovirus known as beta-satellites (~ 1.4 kb) (Briddon and Stanley, 2006, Virology). Betasatellites are believed to have evolved independently and encode a single conserved βC1 protein (Fiallo-Olivé et al., 2021, Journal of General Virology; Saunders, et al., 2000, Proc. Natl. Acad. Sci. USA; Yang et al., 2019, Trends in Plant Science). These results suggest that no suitable roots can be found for constructing the phylogenetic tree of betasatellite species.

(2) We and other groups have shown the similar genomic locus and similar functions, such as long-distance movement protein, silencing suppressor, pathogenicity protein between bipartite begomovirus BV1 proteins encoded by DNA-B component and betasatellite-encoded βC1 proteins (Li et al., 2014, The Plant Cell; Saunders et al., 2002, Virology). More importantly, both BV1 and βC1 interact with MYC2 to suppress plant terpene-based resistance and promote vector performance (Li et al., 2014, The Plant Cell).

(3) Usually, both the DNA-A and DNA-B components are required for the infectivity of bipartite begomoviruses. But in some few cases, DNA-A component of Sri Lankan cassava mosaic virus (SLCMV), Tomato chlorotic mottle virus (ToCMV) and Indian cassava mosaic virus (ICMV) is sufficient for systemic infection and movement in Nicotiana benthamiana (Saunders et al., 2002, Virology; Wang et al., 2014, Virus Genes). Therefore, SLCMV DNA-B has the biological characteristics of a monopartite begomovirus-associated betasatellite and could be replaced by each other (Saunders et al., 2002, Virology; Saeed et al., 2007, Journal of General Virology; Yang et al., 2019, Trends in Plant Science).

(4) Given that SLCMV DNA-B encoded BV1 is very similar to that of betasatellite βC1 proteins, thus SLCMV BV1 is the best choice as an outgroup in the phylogenetic tree for betasatellite species.

Q3: Line 110: ‘Asia-subclade.’?  . is necessary.

Response: Thanks for your critical comments. We have removed this in the revision.  

Q4: Line 739, ‘which lost ’ changes into ‘without’

Response: Thanks for your valuable suggestion. We have changed the words.

Q5: The formats of references could be more consistent. Please double-check all of them. For example, Trends in plant science, PloS one, etc.. are not standard abbreviations for these journals.

Response: We sincerely appreciate the valuable comments. We have checked the formats of references carefully and revised the abbreviations for all cited journals.

Reference

Briddon, R.W; Stanley, J. Subviral agents associated with plant single-stranded DNA viruses. Virology. 2006, 344, 198-210.

Fiallo-Olivé, E.; Lett, J. M.; Martin, D. P.; Roumagnac, P.; Varsani, A.; Zerbini, F. M.; Navas-Castillo, J. ICTV Virus Taxonomy Profile: Geminiviridae 2021. J. Gen. Virol. 2021, 102.

Li, R.; Weldegergis, B. T.; Li, J.; Jung, C.; Qu, J.; Sun, Y.; Qian, H.; Tee, C.; van Loon, J. J. A.; Dicke, M.; et al. Virulence Factors of Geminivirus Interact with MYC2 to Subvert Plant Resistance and Promote Vector Performance. Plant Cell 2014, 26, 4991-5008.

Saunders, K.; Bedford, I.D.; Briddon, R.W.; Markham, P.G.; Wong, S.M.Stanley, J. A unique virus complex causes Ageratum yellow vein disease. Proc Natl Acad Sci U S A. 2000, 97, 6890-6895.

Saunders, K.; Salim, N.; Mali, V.R.; Malathi, V.G.; Briddon, R.; Markham, P.G.Stanley, J. Characterisation of Sri Lankan cassava mosaic virus and Indian cassava mosaic virus: evidence for acquisition of a DNA B component by a monopartite begomovirus. Virology. 2002, 293, 63-74.

Saeed, M.; Zafar, Y.; Randles, J.W.Rezaian, M.A. A monopartite begomovirus-associated DNA beta satellite substitutes for the DNA B of a bipartite begomovirus to permit systemic infection. J Gen Virol. 2007, 88, 2881-2889.

Wang, G.; Sun, Y.; Xu, R.; Qu, J.; Tee, C.; Jiang, X.Ye, J. DNA-A of a highly pathogenic Indian cassava mosaic virus isolated from Jatropha curcas causes symptoms in Nicotiana benthamiana. Virus Genes. 2014, 48, 402-405.

Yang, X.; Guo, W.; Li, F.; Sunter, G.; Zhou, X. Geminivirus-Associated Betasatellites: Exploiting Chinks in the Antiviral Arsenal of Plants. Trends Plant Sci. 2019, 24, 519-529.

Reviewer 2 Report

Begomoviruses and their satellites infect a wide range of crops and weeds around the world. The authors of this study reveal intriguing discoveries about the evolutionary history of Begoviruses and their arthropod-borne vectors. The findings of this study are interesting and will undoubtedly lead to a better understanding of the begomoviruses' successful host-invasion mechanism, how they overcome host resistance, and how their satellites have established a coordinated network of protein interactions.

However, I noticed minor grammatical errors in the introduction section. Some sentences are quite long and difficult to comprehend for the reader. Some grammatical issues are listed below. It is recommended that the authors seek the assistance of an English editing agency or a native English speaker to improve the readability of the document.

Line 46 Plant viruses can directly affect vector behavior.

Line 47 ……with the ingestion [10]

Line 47-48 ……mutualistic relationships between viruses and vectors are….

Line 51-52 …. Understanding the universal agents and what factors manipulate the mutualism between viruses and their ……

Finally, in my opinion. the manuscript can be accepted for publication after minor revision.  

Author Response

Q 1: It is recommended that the authors seek the assistance of an English editing agency or a native English speaker to improve the readability of the document.

Response: Thanks for your invaluable comments. We have polished our language with native English speakers to improve the readability of the manuscript and correct grammar mistakes throughout the entire manuscript.

Q 2: Line 46 Plant viruses can directly affect vector behavior.

Response: Thanks for your critical comments. We have revised the sentences.

Q 3: Line 47 ……with the ingestion [10]

Response: Thanks for your suggestion. We have revised the sentences.

Q 4: Line 47-48 ……mutualistic relationships between viruses and vectors are….

Response: Thanks for your critical comments. We have revised the sentences.

Q 5: Line 51-52 …. Understanding the universal agents and what factors manipulate the mutualism between viruses and their ……

Response: Thanks for your critical comments. We have revised the sentences.

Reviewer 3 Report

This seems to be a thorough study of several beta-C1 proteins and an excellent manuscript. Unfortunately, I have major issues with Figure 1A, which sets the stage for the paper's arguments about conserved function.

One cannot simply “designate” BV1 as the root in the tree. (lines 103 and 122). Using an outgroup sequence would enable rooting the tree, but BV1 does not seem to be an appropriate outgroup sequence. I cannot find any published evidence that beta-C1 genes derive from a BV1 gene, nor do the two genes have any detectable sequence similarity (by BLAST). Therefore including an unrelated BV1 sequence in the tree is probably nonsensical.

The circular phylogeny is compact, but is hard to read (i.e. visually assess distances from the center of the tree). Consider using a linear tree instead. The panel is also pixelated at the current resolution and the bootstrap-confidence-level circles are hard to see on shaded backgrounds.

A list of ten less important issues follows:

title: "begomoviruses evolutionary hijack […] defense" seems ungrammatical to me, like the word ‘evolutionary’ is out of place. The word ‘evolutionarily’ does not work particularly well as an adverb on lines 32 or 34, and is somewhat redundant in the phrase ‘evolutionarily conserved’ (lines 24, 84, 118, 181, 192, 206, 252). 'Diverse begomovirus betasatellites hijack …'?

line 64: 'Betasatellite' should be capitalized. It would be clearer to write "the virus family Tolecusatellitidae, genus Betasatellite", to visually separate the two italicized taxon names.

The first letters of virus names (as distinct from virus species names) should generally not be capitalized, so 'tomato' (not 'Tomato'), cotton, etc. on lines 111, 112, 116, 143. By contrast, I would capitalize ‘Siegesbeckia’ on line 182 (as done elsewhere).

Extra space in 'w hiteflies' in Figure 1B x-axis label.

line 100: I think you mean "conserved trait", not "conservative trait".

line 110: extra period – "subgroup I Asia-subclade. also fell into subgroup I"

line 152 and 160: Referring to N. benthamiana as “tobacco” is not ideal. (Tobacco is N. tabaccum.)

line 320: 'compacity' should be 'capacity', I think.

Figure S1: Consider referring to “relative virus DNA level”, not “relative virus titer” (because infectious units were not determined) and “qPCR” rather that “RT-PCR” (to minimize possible reader confusion about ‘real-time’ vs ‘reverse transcription’ PCR). I do not see a TYLCCNV-alone treatment, so does this figure really show that “betasatellites are required for begomovirus symptoms”?

Figure S5: bottom-most red box is not well-aligned over the TYLCVNB sequence.

Author Response

Q 1: One cannot simply “designate” BV1 as the root in the tree. (lines 103 and 122). Using an outgroup sequence would enable rooting the tree, but BV1 does not seem to be an appropriate outgroup sequence. I cannot find any published evidence that beta-C1 genes derive from a BV1 gene, nor do the two genes have any detectable sequence similarity (by BLAST). Therefore including an unrelated BV1 sequence in the tree is probably nonsensical.

Response: Thanks for your critical comments. We have clarified the reasons for SLCMV BV1 as an outgroup in the revision.

(1) The begomoviruses are known as either monopartite or bipartite begomoviruses according to the number of genomic components. Both the monopartite and bipartite begomoviruses have the DNA-A component. Bipartite begomoviruses contain the DNA-B component. The majority of monopartite begomoviruses are associated with beta-satellites (Briddon and Stanley, 2006, Virology). Many monopartite begomovirus-associated betasatellites could be replaced with bipartite begomovirus, e.g. Sri Lankan cassava mosaic virus (SLCMV) DNA-B (Saunders et al., 2002, Virology; Saeed et al., 2007, Journal of General Virology; Yang et al., 2019, Trends in Plant Science).

(2) Betasatellites are believed to have evolved independently and encode conserved βC1 proteins (Fiallo-Olivé et al., 2021, Journal of General Virology; Saunders, et al., 2000, Proc. Natl. Acad. Sci. USA; Yang et al., 2019, Trends in Plant Science), suggesting that maybe no no suitable roots can be found for constructing the phylogenetic tree

(3) We and other groups have shown that bipartite begomovirus DNA-B encoded BV1 and betasatellite βC1 proteins confer a similar genomic locus and some similar functions, such as long-distance movement protein, silencing suppressor, pathogenicity protein (Li et al., 2014, The Plant Cell; Saunders et al., 2002, Virology). More importantly, both BV1 and βC1 interact with MYC2 to suppress plant terpene-based resistance and promote vector performance (Li et al., 2014, The Plant Cell).

(4) In some cases, especially bipartite begomovirus SLCMV, Tomato chlorotic mottle virus (ToCMV) and Indian cassava mosaic virus (ICMV), their DNA-A component is sufficient to cause symptoms in plants (Saunders et al., 2002, Virology; Wang et al., 2014, Virus Genes). Therefore, SLCMV DNA-B has the biological characteristics of a monopartite begomovirus-associated betasatellite.

(5) Since SLCMV DNA-B encoded BV1 is very similar to that of betasatellite βC1 proteins. Taken together, we consider that SLCMV BV1 is the best choice as an outgroup in the phylogenetic tree.

Q 2: The circular phylogeny is compact, but is hard to read (i.e. visually assess distances from the center of the tree). Consider using a linear tree instead. The panel is also pixelated at the current resolution and the bootstrap-confidence-level circles are hard to see on shaded backgrounds.

Response: Thanks for your valuable suggestion. We have taken your suggestion with a linear tree in the revision.

Q 3: title: "begomoviruses evolutionary hijack […] defense" seems ungrammatical to me, like the word ‘evolutionary’ is out of place. The word ‘evolutionarily’ does not work particularly well as an adverb on lines 32 or 34, and is somewhat redundant in the phrase ‘evolutionarily conserved’ (lines 24, 84, 118, 181, 192, 206, 252). 'Diverse begomovirus betasatellites hijack …'?

Response: Thanks for your suggestion. We have replaced the title with your suggestion.

Q 4: line 64: 'Betasatellite' should be capitalized. It would be clearer to write "the virus family Tolecusatellitidae, genus Betasatellite", to visually separate the two italicized taxon names.

Response: Thanks for your suggestion. We have taken your suggestion and rewritten "the virus family Tolecusatellitidae, genus Betasatellite" in the revision.

Q 5: The first letters of virus names (as distinct from virus species names) should generally not be capitalized, so 'tomato' (not 'Tomato'), cotton, etc. on lines 111, 112, 116, 143. By contrast, I would capitalize ‘Siegesbeckia’ on line 182 (as done elsewhere).

Response: Thanks for your suggestion. We have corrected the spelling in the new version.

Q 6: Extra space in 'w hiteflies' in Figure 1B x-axis label.

Response: Thanks for your correction of the typo. We have deleted the extra space. the spelling in the new version.

Q 7: line 100: I think you mean "conserved trait", not "conservative trait". 

Response: Thanks, we have changed.

Q 8: line 110: extra period – "subgroup I Asia-subclade. also fell into subgroup I"

Response: Thanks, we have changed.

Q 9: line 152 and 160: Referring to N. benthamiana as “tobacco” is not ideal. (Tobacco is N. tabaccum.)

Response: Thanks, we have corrected this in the revision.

Q10: line 320: 'compacity' should be 'capacity', I think.

Response: Thanks, we have corrected the word.

Q11: Figure S1: Consider referring to “relative virus DNA level”, not “relative virus titer” (because infectious units were not determined) and “qPCR” rather that “RT-PCR” (to minimize possible reader confusion about ‘real-time’ vs ‘reverse transcription’ PCR). I do not see a TYLCCNV-alone treatment, so does this figure show that “betasatellites are required for begomovirus symptoms”?

Response: Thanks for your helpful suggestion. We have corrected these words with your suggestion. Previous studies have demonstrated that TYLCCNV-associated betasatellite TYLCCNB is required for symptom induction and mimic phosphorylation of TYLCCNB-eoncoded βC1 impairs disease symptoms (Cui et al., 2004, Journal of Virology; Zhong et al., 2017, Journal of Virology). In fact, we have checked the begomovirus symptoms when only TYLCCNV (TA)-, TA-associated with betasatellite (TA+tβ)-, and TA-associated a βC1 mutant betasatellite (TA+tβS33D) infected N. benthamiana. We found that TA+tβ infection developed disease symptoms with severe curling of leaves and shot twisting, while only TA infection failed to cause symptoms (Figure S1A). The TA+tβS33D infection attenuated disease symptoms. The results suggest that betasatellites are required for begomovirus symptoms.

Q12: Figure S5: bottom-most red box is not well-aligned over the TYLCVNB sequence.

Response: Thanks, we have aligned the TYLCVNB sequence with the red box.

Reference

Briddon, R.W; Stanley, J. Subviral agents associated with plant single-stranded DNA viruses. Virology. 2006, 344, 198-210.

Cui, X.; Tao, X.; Xie, Y.; Fauquet, C.M.Zhou, X. A DNA β associated with Tomato yellow leaf curl China virus is required for symptom induction. J Virol. 2004, 78, 13966-13974.

Fiallo-Olivé, E.; Lett, J. M.; Martin, D. P.; Roumagnac, P.; Varsani, A.; Zerbini, F. M.; Navas-Castillo, J. ICTV Virus Taxonomy Profile: Geminiviridae 2021. J. Gen. Virol. 2021, 102.

Li, R.; Weldegergis, B. T.; Li, J.; Jung, C.; Qu, J.; Sun, Y.; Qian, H.; Tee, C.; van Loon, J. J. A.; Dicke, M.; et al. Virulence Factors of Geminivirus Interact with MYC2 to Subvert Plant Resistance and Promote Vector Performance. Plant Cell 2014, 26, 4991-5008.

Saunders, K.; Bedford, I.D.; Briddon, R.W.; Markham, P.G.; Wong, S.M.Stanley, J. A unique virus complex causes Ageratum yellow vein disease. Proc Natl Acad Sci U S A. 2000, 97, 6890-6895.

Saunders, K.; Salim, N.; Mali, V.R.; Malathi, V.G.; Briddon, R.; Markham, P.G.Stanley, J. Characterisation of Sri Lankan cassava mosaic virus and Indian cassava mosaic virus: evidence for acquisition of a DNA B component by a monopartite begomovirus. Virology. 2002, 293, 63-74.

Saeed, M.; Zafar, Y.; Randles, J.W.Rezaian, M.A. A monopartite begomovirus-associated DNA beta satellite substitutes for the DNA B of a bipartite begomovirus to permit systemic infection. J Gen Virol. 2007, 88, 2881-2889.

Wang, G.; Sun, Y.; Xu, R.; Qu, J.; Tee, C.; Jiang, X.Ye, J. DNA-A of a highly pathogenic Indian cassava mosaic virus isolated from Jatropha curcas causes symptoms in Nicotiana benthamiana. Virus Genes. 2014, 48, 402-405.

Yang, X.; Guo, W.; Li, F.; Sunter, G.; Zhou, X. Geminivirus-Associated Betasatellites: Exploiting Chinks in the Antiviral Arsenal of Plants. Trends Plant Sci. 2019, 24, 519-529.

Zhong, X.; Wang, Z.Q.; Xiao, R.; Cao, L.; Wang, Y.; Xie, Y.Zhou, X. Mimic phosphorylation of a βC1 protein encoded by TYLCCNB impairs its functions as a viral suppressor of RNA silencing and a symptom determinant. J Virol. 2017, 91.

Round 2

Reviewer 3 Report

I must restate my central criticism: Beta-C1 and BV1 are not phylogenetically related, and therefore it is wrong to show them together on a phylogenetic tree. The fact that Beta-C1 and BV1 share molecular and genetic functions is interesting but irrelevant. From the authors' reply, it seems as if they may think that an outgroup sequence is always required to infer a phylogenetic tree, but this is not the case. There is nothing wrong with unrooted trees. As far as I can tell, removing BV1 from the tree will not change any of the conclusions in the manuscript.

Minor: italics in the scientific names Begomovirus and Bemisia tabaci in the abstract (lines 17 and 18) seem to have been lost during revision.

Author Response

Q1: I must restate my central criticism: Beta-C1 and BV1 are not phylogenetically related, and therefore it is wrong to show them together on a phylogenetic tree. The fact that Beta-C1 and BV1 share molecular and genetic functions is interesting but irrelevant. From the authors' reply, it seems as if they may think that an outgroup sequence is always required to infer a phylogenetic tree, but this is not the case. There is nothing wrong with unrooted trees. As far as I can tell, removing BV1 from the tree will not change any of the conclusions in the manuscript.

Response: We really appreciate the critical comments. Since almost half of monopartite begomoviruses are associated with DNA beta-satellites (~ 1.4 kb) (Briddon and Stanley, 2006, Virology). Betasatellites are believed to have evolved independently and encode a single conserved βC1 protein (Fiallo-Olivé et al., 2021, Journal of General Virology; Saunders, et al., 2000, Proc. Natl. Acad. Sci. USA; Yang et al., 2019, Trends in Plant Science). These results indicate that no suitable roots can be found for constructing the phylogenetic tree of betasatellite species. We decided to follow the reviewer’s valuable suggestion. We have removed BV1 from the phylogenetic tree and conducted the phylogenetic analysis with an unrooted tree.

Q2: Minor: italics in the scientific names Begomovirus and Bemisia tabaci in the abstract (lines 17 and 18) seem to have been lost during revision.

Response:  Thanks, we have revised these.

Reference

Briddon, R.W; Stanley, J. Subviral agents associated with plant single-stranded DNA viruses. Virology. 2006, 344, 198-210.

Fiallo-Olivé, E.; Lett, J. M.; Martin, D. P.; Roumagnac, P.; Varsani, A.; Zerbini, F. M.; Navas-Castillo, J. ICTV Virus Taxonomy Profile: Geminiviridae 2021. J. Gen. Virol. 2021, 102.

Saunders, K.; Bedford, I.D.; Briddon, R.W.; Markham, P.G.; Wong, S.M.Stanley, J. A unique virus complex causes Ageratum yellow vein disease. Proc Natl Acad Sci U S A. 2000, 97, 6890-6895.

Yang, X.; Guo, W.; Li, F.; Sunter, G.; Zhou, X. Geminivirus-Associated Betasatellites: Exploiting Chinks in the Antiviral Arsenal of Plants. Trends Plant Sci. 2019, 24, 519-529.

Round 3

Reviewer 3 Report

This revision satisfies my concern (but note that now that SLCMV BV1 node has been covered over the corresponding bootstrap support circle is no longer applicable).

Extra space in Figure 1B x-axis is still present and there may be other small typesetting errors (like a dangling 'n' at the end of line 190).